Sex differences in the use of social information emerge under conditions of risk

Brand Charlotte O.
Brown Gillian R.
Cross Catharine P. cpc2@st-andrews.ac.uk
School of Psychology and Neuroscience, University of St. Andrews , St Andrews , United Kingdom
Hopper Lydia
Electronic publication date: 2018 Jan 3
Publication date: 2018
Volume: 6
Electronic Location ID: e4190
Received 2017 Aug 29; Accepted 2017 Dec 4
Copyright: ©2018 Brand et al.
Copyright year: 2018
Copyright holder: Brand et al.
License: This is an open access article distributed under the terms of the Creative Commons Attribution License, which permits unrestricted use, distribution, reproduction and adaptation in any medium and for any purpose provided that it is properly attributed. For attribution, the original author(s), title, publication source (PeerJ) and either DOI or URL of the article must be cited.
License URL: https://creativecommons.org/licenses/by/4.0/

Keywords: Sex differences, Risk taking, Human behaviour, Social learning, Social information use, Risk aversion, Cultural evolution

Funding: John Templeton Foundation School of Biology, University of St Andrews School of Psychology & Neuroscience, University of St Andrews The research was funded by a John Templeton Foundation grant, awarded to lead principal investigators Kevin Laland (School of Biology, University of St Andrews) and Andrew Whiten (School of Psychology & Neuroscience, University of St Andrews). The funders had no role in study design, data collection and analysis, decision to publish, or preparation of the manuscript.

==============================
Social learning provides an effective route to gaining up-to-date information, particularly when information is costly to obtain asocially. Theoretical work predicts that the willingness to switch between using asocial and social sources of information will vary between individuals according to their risk tolerance. We tested the prediction that, where there are sex differences in risk tolerance, altering the variance of the payoffs of using asocial and social information differentially influences the probability of social information use by sex. In a computer-based task that involved building a virtual spaceship, men and women (N = 88) were given the option of using either asocial or social sources of information to improve their performance. When the asocial option was risky (i.e., the participant’s score could markedly increase or decrease) and the social option was safe (i.e., their score could slightly increase or remain the same), women, but not men, were more likely to use the social option than the asocial option. In all other conditions, both women and men preferentially used the asocial option to a similar degree. We therefore found both a sex difference in risk aversion and a sex difference in the preference for social information when relying on asocial information was risky, consistent with the hypothesis that levels of risk-aversion influence the use of social information.

Introduction

Individuals can acquire information either directly through their own asocial learning experiences or by copying other individuals (Hoppitt & Laland, 2013). Asocial learning allows individuals to gain first-hand knowledge about the immediate environment, but reliance on this type of learning can be costly, for instance, in terms of time and energy (Kendal et al., 2005). In contrast, social learning can provide a cost-effective route to gaining up-to-date information, particularly when the environment is changing and information is costly to obtain asocially (Kendal et al., 2005; Boyd & Richerson, 1985). Theoretical models support the hypothesis that an increased reliance on social learning is adaptive when the environment becomes more variable (although not when variability is very high) and when the returns from asocial learning become more unreliable (Boyd & Richerson, 1985; Arbilly et al., 2011; Boyd & Richerson, 1988; Feldman, Aoki & Kumm, 1996). Therefore, individuals are predicted to be sensitive to the reliability of the available sources of information and to use these reliability estimates when choosing whether to learn asocially or socially (Kendal et al., 2005).

Reliability can include the predictability of the source of information (e.g., the likelihood that a food reward is associated with a particular cue) and the variability in the expected payoff derived from different sources (e.g., the variability in the amounts of food obtained from different foraging patches). Empirical research on non-human animals and humans has shown that individuals are likely to use social learning when personal experience reveals that the environment is unpredictable or the variability in payoffs of available options is high (e.g., Çelen & Hyndman, 2012; Jones et al., 2013; Rafacz & Templeton, 2003; Van Bergen, Coolen & Laland, 2004). For example, a study of nine-spined sticklebacks (Pungitius pungitius) showed that reducing the predictability of personally-experienced cues in a foraging context increased reliance on social learning (Van Bergen, Coolen & Laland, 2004). Similarly, when faced with the option of a risky or safe action in an experimental paradigm, human participants were found to delay their decision and observe the choices made by others, especially if their private information did not support the risky action (Çelen & Hyndman, 2012). These findings support the broader hypothesis that social learning is used strategically (Laland, 2004).

Individual differences in the use of social information are predicted to reflect individual differences in risk tolerance (Webster & Ward, 2011). Risk-averse individuals are expected to switch to using social sources sooner than risk-prone individuals when faced with unreliable personal experience. In real-world scenarios, the predictability and riskiness of sources of information are likely to co-vary; for example, food items with high nutritional value are likely to be both rarer in the environment, and more difficult to obtain, than low value food items (Arbilly et al., 2011; Brown, Almond & Van Bergen, 2004). By switching to social sources of information when faced with risky options, individuals are thus potentially better able to exploit high-value resources. In both non-human animals and humans, individuals vary in their sensitivity to experiencing gains and losses (Dohman et al., 2011; Reale et al., 2007), and a small number of studies of non-human animals have revealed that ‘shy’ individuals are more likely than ‘bold’ individuals to copy the behaviour of others (e.g., Carter et al., 2013; Harcourt et al., 2009; Kurvers et al., 2010; Kurvers et al., 2011). However, the link between risk-proneness and social information use has yet to be evaluated in humans.

One variable that is commonly related to risk aversion in humans is an individual’s sex, with women obtaining lower average scores than men on a range of risk-taking measures (e.g., Byrnes, Miller & Schafer, 1999; Charness & Gneezy, 2012; Cross, Cyrenne & Brown, 2013). While the degree of overlap between the sexes on risk-aversion measures is often considerable (Nelson, 2015), and not all risk-aversion measures show sex differences (Harris, Jenkings & Glaser, 2006), women perceive the benefits gained from taking risks as being lower than do men (Harris, Jenkings & Glaser, 2006; Weber, Blais & Betz, 2002). Women also rate both the likelihood of a negative outcome and the perceived severity of the costs higher than do men (Harris, Jenkings & Glaser, 2006; Weber, Blais & Betz, 2002), and report being less likely than men to engage in novel activities that involve risk (Cross, Cyrenne & Brown, 2013). Similarly, data from personality measures indicate that, on average, women are more sensitive than men to the potential negative outcomes of decisions (Cross, Copping & Campbell, 2011). The probability of using asocial versus social sources of information when faced with a risky decision is therefore likely to differ on average between women and men.

The aim of this study was to examine whether altering the riskiness of using asocial and social sources of information would differentially influence the probability that men and women used these sources. Here, we are defining riskiness in terms of variation in expected score (Cross, Copping & Campbell, 2011). We predicted that, when one of these sources of information appeared to be risky (i.e., high variation in expected score), women would be more likely than men to use the alternative source, safe (i.e., low variation in expected score) of information. In the control condition, no sex difference in the use of asocial and social sources was predicted. We designed a novel computer-based task that involved constructing a virtual spaceship. After building the first spaceship, participants were given the option of using asocial or social sources of information to improve their ranked score. Participants were assigned to one of three conditions, in which either (i) the asocial option was risky and the social option was safe, (ii) the asocial option was safe and the social option was risky, or (iii) both the asocial and social options were safe. Because scores were randomly allocated to spaceships, participants could not learn about the usefulness of different design features. The outcome measure of principal interest was the participant’s choice of information source. Participants also completed a risky impulsivity measure (Campbell & Muncer, 2009), in order to confirm that the predicted sex difference in average score was found in our set of participants.

Methods

Participants

Eighty-eight participants (50 women and 38 men) were recruited through the University of St Andrews’ School of Psychology & Neuroscience online participant recruitment system. All participants were aged 17 or over, with the majority (91%) falling into the 18–25 age range. Participants gave consent before taking part in the experiment and were debriefed afterwards. All participants were reimbursed £3 for attending the session—which lasted approximately 20 min—and could obtain an additional £2 depending on performance criteria (see ‘Procedure’ below). Participants were randomly assigned to one of the three conditions (see ‘Asocial and social information’) and were tested in groups (range = 4–9 individuals). Participants gave consent via a button click at the start of the experiment. This was approved, as were all other procedures used in this study, by the Ethics Committee of the School of Psychology & Neuroscience on behalf of the University of St Andrews (approval code PS11481).

Procedure

Participants stated their gender (‘female’, ‘male’), age bracket, current level of education and country of origin before beginning the experiment. They then played a computer game, programmed using web-based JavaScript, in which they built virtual spaceships. Participants were instructed that the aim was to construct spaceships with the highest scores, and that the participant with the highest score at the end of the session would receive a bonus payment. Spaceship construction proceeded in three rounds, each with two building phases. In Phase 1, participants constructed their first spaceship by selecting tiles from a grid of thirty available items that were arranged into themes (crew, cargo, engines, shields and lasers) (see Fig. 1). Players had two minutes to view these items and choose ten to place on a spaceship template. The only constraint was that they had to use at least one crew member and one engine. After finishing Phase 1, each player’s ship was given a numerical score and a rank in a league table (1st to 5th, highest to lowest) (see Fig. 2). Players were given no information on how a good score might be achieved, and, in reality, scores were randomly assigned to the participants’ spaceship (with a range of 8,000 to 25,000), along with a false ‘rank’ that was always either 1st, 3rd or 5th.

Figure 1 Screenshot of Phase 1.

Example screenshot from the online experiment showing shipbuilding in Phase 1.

Figure 2 Screenshot of Phase 1.

Example screenshot of the generated score presented to participants at the end of Phase 1.

Participants then chose between using asocial and social sources of information (see ‘Asocial and social options’) (see Fig. 3) before building a second spaceship (Phase 2). Participants were not given a score or rank for their second spaceship at the end of Phase 2, meaning that they received no feedback on whether the choice to use asocial or social source of information improved the outcome. Furthermore, because scores were randomly generated, no rules for building high-scoring spaceships were available for participants to learn.

Figure 3 Screenshot of the choice participants saw.

Screenshot of the two choices participants saw in the two conditions. The top two choices were displayed in the Risky Social condition, the bottom two were displayed in the Risky Asocial condition. Which option was displayed on the right or left was randomised in both conditions.

Phases 1 and 2 were then repeated a further two times (i.e., three Rounds of building spaceships), with scores and ranks shown at the end of each Phase 1. Each participants’ spaceships were ranked randomly, once at 1st, 3rd or 5th. These rankings were displayed on a league table to the participants. At the end of each Round, participants were informed that their score had been saved and that their best score out of the three Rounds would be used at the end of the experiment to allocate the bonus. Because the scores given to spaceships were random, bonuses were awarded at the end of the experiment to more than one participant according to a lottery in which participants had a higher probability of a reward when choosing the safe rather than the risky option in the final Round.

After completing all three Rounds, participants completed the 12-item risky impulsivity measure (Campbell & Muncer, 2009) on-screen. This measure assesses willingness to take risks without prior thought in everyday life and is reported to have high internal consistency (Campbell & Muncer, 2009). The bonus payments were awarded when all participants had completed the final on-screen material, including the questionnaire.

Asocial and social options

The asocial option consisted of viewing up to ten previously unseen items in the scrapheap, of which up to three items could be kept for use in the next building Phase. The social option consisted of viewing three completed spaceships, ostensibly built by ‘other participants’, along with the associated ‘scores’. These spaceship designs had actually been generated by the experimenter prior to the study using randomly selected tiles, and three out of twelve completed spaceships were presented at random as social sources. The scores for these spaceships were also randomly assigned. Participants could choose up to three items from one of the three ships, and these items were automatically added to the participant’s spaceship template in the next Phase and could not be removed. Because each participant was assigned to a single condition, the description of the asocial and social options (see below) remained the same for participants across Rounds, in order to avoid potential confusion among participants and reduce the chance that participants failed to attend to the subtle differences in the descriptive material.

In the Asocial Risky (AR) condition (N = 28; 18 female, 10 male), participants were informed that their score could markedly increase or decrease if they visited the scrapheap, in a short paragraph that included the following wording: “some of these items may be broken and useless, but some may greatly increase your ship’s score…your score could go up or down”. Conversely, the social option was safe; participants were informed their score could slightly increase or would remain the same (“the ships will have the same score as your ship, or slightly higher…you will be guaranteed at least the same score as your current ship”). In the Social Risky (SR) condition (N = 32; 17 female, 15 male), the asocial option was safe (“all of these items will help your ship to fly, and some of them can slightly increase your ship’s score…you will be guaranteed at least the same score as your current ship”), and the social option was risky (“the ships may have a much worse or much better score that your current ship’s score…your score could go up or down”). In the Control (C) condition (N = 28; 15 female, 13 male), both the asocial and social options were safe: the wording was identical to that used in the safe options in the other conditions. This wording reflects the Bounded Risk Distribution model, in which individuals are expected to maximize their probability of reaching a goal while minimizing their probability of falling below a certain threshold (Wang, 2002). In our experiment, participants are trying to achieve the goal of a top score and want to minimize their chance of falling below this threshold, in order to achieve a bonus payment. Therefore, although the safe options have a slightly higher average expected score, these safe options preclude a large increase in score. The risky option is therefore a rational choice where participants believe that they need to greatly improve their score in order to move up in the rankings and win a monetary bonus.

Statistical analyses

We modelled the participants’ decision to use asocial or social options using Bayesian binomial multi-level logistic regression in R with the map2stan function from the Rethinking package (McElreath, 2016). The full model included an effect for sex, an effect for condition, a sex*condition interaction, an effect for the rank given to the participant’s spaceship after Phase 1, and a random effect for individual. The C condition was represented as the baseline in the model, so that any effects of the AR or SR conditions were in relation to C. Because men were coded as 0 and women as 1, the baseline represents men’s behaviour in the control condition, and the effect of sex represents how women’s behaviour differed from men’s in the C condition. Model predictions were calculated by averaging across all candidate models weighted according to the WAIC (Watanabe-Akaike Information Criteria). The model with the lowest WAIC value, and the highest Akaike weight, is the model that is most likely to make accurate predictions on new data, conditional on the set of models considered. Posterior predictions were calculated based on the population mean of the participants and thus represent predictions for a ’new’, previously unobserved, average participant. These predictions are presented in Fig. 4. Candidate models were chosen based on a priori hypotheses formulated before data collection (Table 1).

Figure 4 Model predictions.

Model predictions of the mean proportion of individuals that chose social information plotted according to condition and sex. Predictions were averaged across all models and weighted according to WAIC weight. Error bars show 89% CIs. Raw means are also plotted, represented by a cross symbol.

Table 1 Candidate models and their WAIC weights.

List of candidate models that were included in the asocial/social information analysis, the a priori hypotheses and the included parameters, with model values (WAIC ± SE) and weights (Akaike weight). Bold type indicates the best fitting model.

Model	Hypothesis	Parameters included	WAIC (±SE)	Akaike weight	
1	Null	Intercept	360.3(5.10)	0.00	
2	Full	Intercept + sex + AR + SR + sex*AR + sex*SR + rank + personality	357.1(10.36)	0.01	
3	Sex and condition interactions predict choice	Intercept + sex*AR + sex*SR	351.1(9.11)	0.25	
4	Sex, and sex and condition interactions, predict choice	Intercept + sex + sex*AR + sex*SR	349.6 (9.77)	0.52	
5	Sex and condition predict choice	Intercept + sex + AR + SR	354(8.92)	0.06	
6	Only condition predicts choice	Intercept + AR + SR	352(8.77)	0.16	
7	Only sex predicts choice	Intercept + sex	362.3(5.32)	0.00	

In order to examine whether the choice of using risky or safe options varied with the rank assigned to the spaceship, or sex of participant, and whether men and women responded differently to their rank assignments, we ran an additional model with risky/safe choice rather than social/asocial choice as the outcome variable. This model excluded data from the C condition, because both options in this condition were safe and therefore no risky choice could be made. The risky/safe choice was modelled using a Bayesian binomial multi-level logistic regression with rank, sex and a sex*rank interaction as predictors.

Finally, we also modelled participants’ risky impulsivity scores using a Bayesian linear model, with sex as a predictor variable, to check whether our sample displayed the expected sex difference in risky impulsivity.

All model estimates are reported with 89% credible intervals (CIs), which are the default in the Rethinking package (McElreath, 2016). The CIs provide an upper and lower estimate around the mean of the parameter estimate and encompass 89% of the posterior distribution. This method contrasts with the traditional use of 95% confidence intervals in null hypothesis testing. Using 95% intervals would not change the interpretation of our results, because we are using a model comparison approach, and the size of the credible intervals does not affect which models best fit the data. All error bars are 89% credible intervals and can be interpreted as the region within which the model expects to find 89% of responses, given the data and the assumptions in the model.

Figure 5 Plot of parameter estimates of best fitting model (lowest WAIC).

Plot displaying parameter estimates for the probability of choosing the social option, taken from the model with the lowest WAIC value and plotted with 89% CIs. A positive estimate indicates a greater likelihood of choosing social, rather than asocial, information. Where the 89% CIs of parameter estimates include zero, there is no clear evidence of an effect of that parameter on the likelihood of choosing the social or asocial option. The intercept (baseline) represents males in the control condition.

Results

Asocial versus social options model

When modelling the probability of choosing asocial or social options, the best-fitting model (i.e., the model with the lowest WAIC value) included an effect for sex and an effect for the sex and AR condition interaction (Table 1). This interaction can be seen in detail in Fig. 4. In the C condition, both women and men preferentially chose to use the asocial source information rather than the social source (women: β =  − 0.72, CI [−1.36, −0.05]; men: β =  − 0.41, CI [−0.72, −0.10]; Fig. 4). As shown by the model estimates (Figs. 5 and 6), there was no strong evidence for an interaction effect between sex and SR condition (β = 0.62, CI [−0.09–1.40]), meaning that women’s choices in the SR condition did not differ strongly from women’s choices in the C condition. Thus, as can be seen in the model predictions (Fig. 4), both women and men preferentially chose the asocial source in the SR condition also. In contrast, the interaction between sex and the AR condition had a strong effect in the model (β = 1.76, CI [1.03–2.51]; Fig. 5). As can be seen in Fig. 5, women in the AR condition preferentially chose the social option, whereas men’s choice did not differ compared to men’s choices in the control condition. Thus, women in the AR condition chose the social source of information more than women in the C condition, while men in the AR condition did not differ from men in the C condition with regard to their choice. According to the full model (Fig. 6), rank did not predict the choice to use asocial or social options (β = 0.05, CI [−0.22–0.29]).

Figure 6 Plot of parameter estimates from the Full Model.

Plot displaying parameter estimates for the probability of choosing the social option, taken from the full model and plotted with 89% CIs.

Figure 7 Plot of parameter estimates from the second model.

Plot displaying parameter estimates for the likelihood of choosing the risky option, plotted with 89% CIs. A positive estimate indicates a greater likelihood of choosing the risky, rather than the safe, option. Where estimates include zero, there is no clear evidence of that parameter affecting the likelihood of choosing the risky or safe option.

Risky versus safe model

The risky versus safe model indicated that participants of both sexes preferred to use the safe option overall (β =  − 2.38, CI [−4.20, −0.56]; Fig. 7). The intercept estimates the preferences of men in both conditions, showing that they had an overall preference for the safe choice, and the effect of sex included zero (β = 1.12, CI [−0.71–3.13]), indicating that women did not choose differently from men. However, the effect of rank (β = 1.14, CI [0.22–1.95]) shows that both men and women were more likely to choose risky than safe options after receiving a lower rank than a higher rank. There was no evidence for an interaction between rank and sex in the model (β =  − 0.77, CI [−1.98–0.19]), indicating that men and women were responding similarly to their rank assignments.

Risky impulsivity measure

Women scored lower than men on the risky impulsivity measure, as expected (women = 23.41 ± 6.97; men = 27.24 ± 7.59; means and SEMs) (β =  − 0.06, CI [−0.09, −0.04]; Fig. 8). Men scored half a standard deviation higher than women on average (Cohen’s d = 0.52). The scale had an acceptable level of internal consistency (Cronbach’s alpha = 0.78). Individual scores did not predict the use of risky versus safe options (β = 0.13, CI [−1.75–1.90]).

Figure 8 Density plot of sex difference in risk-taking measure.

Density plot showing men and women’s risky impulsivity scores.

Discussion

In our experimental study, we found a sex difference in the choice to use social information that only emerged when the alternative, asocial option was risky. Women, but not men, preferentially chose to use the social source of information when the asocial option was risky. In contrast, women and men did not differ from each other in their responses to risky social options; both women and men preferentially used the asocial option in the ‘Social Risky’ condition, as well as in the control condition. Male and female participants were more likely to choose the risky option when the spaceship was given a low rank than a high rank, while rank did not predict whether participants chose asocial or social information. Women had lower average scores than men on a risky impulsivity measure, as reported in previous research (e.g., Cross, 2010; Campbell & Muncer, 2009). Our main finding, which was that individuals of the more risk-averse sex (i.e., women) used the social option when the asocial one was risky, is consistent with the hypothesis that levels of risk-aversion influence social learning strategies (Arbilly et al., 2011). This result has potentially broad implications for understanding the dynamics of social information transmission.

While previous research has suggested that women are more likely than men to conform to the decisions of others (e.g., Bond & Smith, 1996), our findings contribute further evidence that social sources of information are used strategically, by both men and women. We found that the sex difference in the use of social sources of information depended upon the type of decision being made. Women were not more likely than men to use social options across all conditions, nor were women less likely than men to choose the risky option in general. The sex difference in the use of the social sources of information when the asocial source was risky could potentially have reflected lower confidence in one’s own performance in women compared to men. Previous research has shown that both female and male participants copy others when lacking confidence in their personal information (e.g. Morgan et al., 2012), and that this relationship is likely to influence patterns of conformity in cases where men’s and women’s confidence differs (Cross et al., 2017). However, the absence of a sex difference in the control condition suggests that both sexes were equally confident in solving the task alone.

The psychological mechanisms underpinning the sex difference in response to risky asocial sources remain to be determined. While a sex difference in competitiveness has been identified in previous literature (Gneezy, Niederle & Rustichini, 2003), this sex difference is unlikely to explain our results because men and women responded similarly to their Phase 1 ranks. One possible explanation is that women were more sensitive on average than men to the potential loss in score associated with the risky asocial option and were thereby minimizing their probability of a loss. However, women and men did not differ in their probability of selecting a risky social option, possibly because they had a preference for the asocial option irrespective of risk. Neither female nor male participants avoided the social option completely, even when it was risky. Participants might have been sampling the social sources in order to compare their own decisions with those of others or to check for particularly high-scoring solutions. This sampling strategy might have prevented participants from relying solely on the asocial option in the social risky condition, which might have resulted in a ceiling effect. Altering the experimental design to make the social option more appealing (in terms of perceived benefits gained from viewing social sources) might have reduced overall reliance on asocial sources when this social information became risky.

Our results showed that both men and women used asocial, rather than social, sources of information when both sources were safe, consistent with previous experiments showing the preferential use of asocial learning in laboratory settings (e.g., Morgan et al., 2012; Mesoudi, 2011). While theoretical models have suggested that social learning should initially be prioritised over asocial learning (Enquist, Eriksson & Ghirlanda, 2007), our empirical research suggests that participants prefer to try to solve tasks for themselves, before relying on help from others. The asocial version of our task, which involved viewing new tiles in a scrapheap, could have been more appealing than the social condition, in terms of providing opportunities to innovate or for other reasons related to the characteristics of the stimuli. Although the probability that men chose the social option did not vary across conditions, the level of risk could potentially influence use of social sources of information by men under different experimental conditions. For instance, further increasing the riskiness of the asocial option could potentially result in men switching to using the social option. The idea that men and women could differ in the cut-off point at which the risk is deemed sufficiently high to change strategy could be investigated experimentally by varying the level of risk along a continuum.

Our results confirmed that participants of both sexes were more likely to choose the risky option when the spaceship was given a low rank than a high rank. Previous experiments using economic game protocols have also shown that participants are more likely to take a risk when performing poorly in relation to other participants (e.g., Atkisson, O’Brien & Mesoudi, 2012; Kahneman & Tversky, 2013). Given that our definition of riskiness focused on the variation in expected score, rather than the absolute size of the expected score, future studies could manipulate both average scores and variance in scores of different options to examine the influence of these on the strategic use of social information in men and women. In the current study, the average score for the safe option was slightly higher than for the risky option, given that scores in the safe option could either increase slightly or remain stable, while scores in the risky option could either increase or decrease markedly. The effects of manipulating level of risk and differences in average payoffs could be further investigated experimentally in both human beings and non-human animals, using available protocols (e.g., Çelen & Hyndman, 2012; Van Bergen, Coolen & Laland, 2004; Kurvers et al., 2011).

Individual scores on the risky impulsivity measure did not correlate with the likelihood of choosing the risky versus safe option. While this correlation was not the main focus of the study, one suggestion for future research might be to examine measures of sensitivity to ‘actuarial’ risk, although these measures are less likely to show sex differences than measures of sensitivity to physical risk (Byrnes, Miller & Schafer, 1999). Indeed, we chose risky impulsivity because we were looking for a trait that differs by sex. A second possibility is as follows. In our task, choosing the ‘risky’ option is rational when a large increase in score is needed, but it brings with it the possibility of a large ‘loss’ in score. We could consider a decrease in score when selecting the rational option as a form of unrepresentative negative feedback (Toelch et al., 2011, see also Cross, Copping & Campbell, 2011), to which women appear to be more sensitive than men. We might therefore expect sensitivity to negative feedback in, for example, a gambling task to correlate with a shift in strategy in our spaceship-building task. Consistent with this explanation is the idea that women may be more sensitive to “social” punishment, in this case by viewing their decline in a league table with others. Studies of different domains of risk or punishment sensitivity would be welcome in order to explore these hypotheses further.

Conclusion

Our results indicated that individuals of the more risk-averse sex preferentially used a social option when the asocial option was risky, supporting theoretical evidence that levels of risk-aversion are linked to the implementation of social learning strategies (Arbilly et al., 2011). Whether the psychological mechanisms underpinning the decision to use social sources of information involved greater sensitivity to punishment or lower confidence in one’s own performance was not investigated. However, regardless of the mechanism, switching to social learning can potentially provide individuals with the opportunity to avoid costly mistakes and learn from the successes of others. Understanding how sex differences in risk-aversion relate to the use of social information deserves further investigation in non-human animals, as well as humans, and would add to the growing evidence that individual traits influence a broad range of social processes (Webster & Ward, 2011; Mesoudi et al., 2016). Between-individual differences in risk-aversion are likely to influence the dynamics of social learning and the spread of socially transmitted information through populations, with potential broad-scale implications for the characteristics of local traditions and the evolution of cultural traits.

Supplemental Information

Data S1 Raw data

Click here for additional data file.

We are grateful to Ken Munro for assistance with programming and to Kevin Laland and Mike Webster for comments on the manuscript.

Additional Information and Declarations

Competing Interests

Author Contributions

Human Ethics

Data Availability

The authors declare there are no competing interests.

Charlotte O. Brand conceived and designed the experiments, performed the experiments, analyzed the data, contributed reagents/materials/analysis tools, wrote the paper, prepared figures and/or tables, reviewed drafts of the paper.

Gillian R. Brown and Catharine P. Cross conceived and designed the experiments, wrote the paper, reviewed drafts of the paper.

The following information was supplied relating to ethical approvals (i.e., approving body and any reference numbers):

The University of St Andrews granted ethical approval to carry out this study.

The following information was supplied regarding data availability:

Github: https://github.com/lottybrand22/GH_RiskySpaceships/.

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
