# Peer review of "Sex differences in the use of social information emerge under conditions of risk"

_PeerJ, doi:10.7717/peerj.4190_

## Round 0.1 · original submission · Minor Revisions

Three experts have now reviewed your article and each has identified merits in your methodology. However, all three reviewers raise some concerns both with regard to your experimental protocol and your interpretation of your results.

Reviewer 1 highlights a potential reinterpretation of your results in terms of participants’ reward maximizing behavior, counter to risk avoidance, and I encourage you to address this. Reviewer 1 also noted your interchangeable use of “gender” and “sex” throughout your article. Please be consistent and, as suggested by reviewer 1, if the participants self-identified, then I would recommend using the term gender rather than sex.

Furthermore, both reviewers 1 and 2 note the interplay between gender and use of social information and I agree with them that unpacking this finding more fully would generate a more thoughtful and thorough article.

Reviewer 3 has requested greater clarity in the presentation of your experimental conditions and I agree that this would be useful. I would also ask that you provide the full text that was presented to participants in each of the conditions, rather than the truncated version you currently cite. Reviewer 3 also raised concerns about the (unintentional) differences generated by the social and asocial conditions in terms of cognitive complexity; how such variability may have impacted the behavior of your participants, merits discussion in your article. Finally, reviewer 3 also challenges your use of deception. Although myself a psychologist and used to human subjects testing often involving deception, I am unsure as to its purpose in these methods. Indeed, it seems to only increase the risk for participants across all conditions. Surely reliable feedback, especially in the safe conditions would have been more valid. Please explain your reasoning for your payment system here and also the random generation of scores.


In addition to the three reviewers’ very detailed and thoughtful feedback, I have a few more minor comments that I request you address:

Line 60 “reducing the predictability of personally experienced cues” should be “reducing the predictability of personally-experienced cues”

Line 62 where you state “delay their decision and observe the choices made by others”, to what is the comparison discussed here? Were participants delaying their decision making in comparison to participants presented with a less-risky decision?

Line 75 “behavior” might be a better word here than “decisions”

Line 78 where you state “lower average scores” please clarify what this translates to. I assume you mean that women are more risk averse than men, but clearly stating this would be helpful here.

Line 108, please provide the university and country where participants were recruited, not just the department. Additionally, do you have any more detailed information regarding the demographics and socioeconomic background of your participants; I am assuming that culture likely interplays with the gender differences reported for risk aversion.

Line 188 there is an extraneous “.” at the end of this line.
In your results section, rather than using acronyms for your conditions (e.g., C, AR), I think it would be more clear if you used the full terms so that the reader can easily track the comparisons.

·

Basic reporting

Generally the basic reporting is good. A few issues below:
- You use sex and gender interchangeable. Sex is biological, gender is cultural. See for example line 77 compared to line 121 and line 345. If you let people self-identify, I suspect you should use gender throughout rather than sex.
-Along this same line, you note that you had four options for participants to select for gender including other and prefer not to say. Did anyone select these options? If so, were they excluded from the analyses? Please specify.
- Line 116 Typo: of not f
-Line 143: You haven't used the term league table before. Perhaps reword to standings list or something like that.
-Line 204: I believe you mean Figure 3 in this line.
-Line 306 Typo, a period, at the beginning of the line.

Experimental design

The methods themselves were fine. However, I have questions about how they relate to the research question. See comments in the validity section.

Validity of the findings

I am unclear why you decided to make this a competitive task. It seems like risk preferences may well vary when in a competitive situation versus other situations. I would like to see some information about how being a competitive task or not influences risk preference. In other words, do competitive situations change risk tolerance?

My biggest issue is that you are confounding risk preference and score maximization. This is an issue that many studies examining risk preferences have. For example, in the asocial risky condition, since the asocial option is more risky, the social option will lead to higher rewards, since it has a higher expected value. Thus, women may be maximizing their score rather than interacting with risk, per se. In the other conditions, their performance is also consistent with reward maximization. Therefore, an alternate interpretation of your findings is that women are more sensitive to rewards (not punishments as you say in line 349) or that they are more willing to switch strategies when it is optimal to do so. Thus, I find your explanations for this behavior (punishment and lack of confidence) quite unsatisfactory. What I think is perhaps the most interesting is that men continue to avoid social information even when doing so leads to higher scores. So, men are not maximizing their rewards in the asocial risky condition. This suggests to me that men my be more averse to social information than women.

Because of the conflation or risk and score maximization, I think what you actually tested is willingness to use social information. You show in your control condition that in labs people avoid social information (this is also supported by the literature). Then, in your social risky condition you show that people continue to avoid social information even when there is a potential, but not guaranteed, benefit. Finally, you show in the asocial risk condition that women, but less so men, are willing to use social information when it is beneficial.

I am not sure how you can disentangle risk preferences and score maximization with your existing data. Perhaps there is another control condition you could add, but I am having trouble imagining what that would be. Alternatively, you could restructure the paper to focus on preferences for asocial/social information across various payout schedules.

Reviewer 2 ·

Basic reporting

The article is well-written and nicely presented. I found it very clear and easy to follow. The literature review places the work suitably in context.

The raw data file was easy to access and understand. I found that the column headers did not line up with the data when I opened the file in Excel (the column headers were misaligned by one cell, relative to where they should have been), but this may have been a quirk of using Excel to open a txt file which might otherwise have displayed correctly in some alternative package. However, perhaps the authors can take a quick look at this and check that everything displays correctly in the most commonly used software packages that they would expect readers to be using.

The figures were reasonably helpful. Having looked at the response to reviewers' comments for the previous submission, I think the decision to include the "real" means in one of the figures was prudent, and actually these do potentially cast a slightly different light on the results, as I will elaborate in Section 3.

Experimental design

The experimental design is essentially very simple. The key manipulation was that participants were told that one option available to them was risky (operationalised as potentially reducing score, but also potentially significantly increasing it), and the other safe (operationalised as guaranteed not to reduce to score, but with only marginal increases possible). These possibilities were counterbalanced across the option types ("social" operationalised as the ostensive attempt of another participant, and "asocial" operationalised as an alternative source of solution elements not presented as having originated from another's attempt). In a control condition both sources were presented as being safe. The measure of interest is simply the choice made by the participant (which could be variously: social safe over asocial safe; social risky over asocial safe; social safe over asocial risky).

In spite its simplicity (and I regard simplicity as compliment rather than criticism!), it is clear from the manuscript how the manipulation makes a novel contribution, in looking at how gender differences in risk sensitivity might interact with increased reliance on social sources of information under conditions of risk. The design is appropriate to tackle this question.

From the information provided, the study appears to have been run with due attention to methodological rigour, and the methods are described in sufficient detail.

Validity of the findings

The authors' conclusion (that female participants are particularly likely to rely on social sources of information when asocial sources are known to be more risky) appears to be upheld by the data.

My only concern here is that I suspect that this might present a somewhat incomplete picture of the results. There are a couple of things that I think the authors might want to consider in order to present a more fleshed out interpretation. I do want to emphasise however that this would entail in one case a speculative interpretation, and in another case, a post-hoc analysis of results. So if the authors take either of these suggestions on board then of course it is important that they make these caveats clear. In addition, given the need for the caveats, I think it is fine if the authors prefer not to stray into this territory at all. However, for me, it would make for a slightly more thought-provoking paper, and one that might be more likely to generate further literature investigating these possibilities in a more a priori fashion.

The first suggestion concerns my point about the "real" means in the Figure. I'm really glad the authors included these. Perhaps it makes me very old-school but I much prefer to see real data over the outputs of models. And in this instance I think that it is particularly helpful as these seem to suggest that something quite interesting might be going on which is not captured by the models. It looks like in the control condition (safe, safe), female participants are actually LESS likely than males to opt for the social source of information. This seems quite interesting, and not what one would expect based on general assumptions and theories about gender and social learning? And then in the social risky condition, women actually appear to INCREASE their tendency to select the social source. This is really interesting too, and certainly it seems consistent with the authors' interpretation that female participants are not simply more risk-avoidant. The pattern of results suggests that actually female participants may respond to ANY cues to potential risk by increasing reliance on information from social sources (even when it is that very social source that has been identified as risky). Of course, I may be completely wrong, and this might not be upheld in the data at all. But it looks intriguing, and therefore potentially worth a bit of post-hoc analysis (explicitly flagged as such, of course) to see whether there might be anything in this interpretation.

And regarding the interpretation, the authors state quite clearly (and correctly, based on their data), that the female participants are not just more risk-averse. And indeed, in line with this, the effect appears not to be explained by risk aversion anyway, based on the lack of relationship with the personality measures. However, the authors do little to propose an alternative interpretation, and I think this would be merited. What exactly IS driving the effect, if it is not simply risk sensitivity? Personally I would like to see the authors propose an alternative interpretation which might be testable in future research. My hunch would be that female participants are particularly sensitive to something that might be labelled as social shame, or similar. If one turns out to have made a choice that sets one apart from the crowd in a "good" way, then there's no shame. But if one underperforms relative to the crowd then perhaps the sense of having foolishly deviated from established behaviour is possibly more aversive to women than to men. This might even explain why women appear to be slightly more inclined to use the social information even when it has been identified as risky, relative to the control condition where it is identified as safe. It might be the case that the mere possibility of one's score going down at all leads women to rely more heavily on social information (since it is not so much the possibility of having made an adverse decision that is disturbing, as the possibility of being the only one to have done so - and copying another's design guarantees that this will not be the case).

Anyway, as previously indicated, these are merely suggestions. I leave it up to the authors whether they take them on board (here or elsewhere). I have no problem with the very neutral interpretation as currently presented.

·

Basic reporting

1. The study presents an experiment exploring the relationships between risk preference and preferences over social or asocial information sources. As such present tense is appropriate. The use of past tense is unnecessary. For example line 108 could be rewritten as: Eighty eight participants are recruited, … give consent… are reimbursed…” etc. I checked several experimental papers on my desk and all follow this convention. Although this is a relatively minor point.

2. Given that the payments are small and deceptive I suspect that wording is very important. I would have found a box listing the choices in full useful. Something like the following but with the actual wording given to subjects.

Conditions

Asocial information source “consisted of viewing up to 10 previously unseen items in the scrapheap, of which up to three items could be kept for use in the next building Phase.”

Social information source: “viewing 3 completed space ships ostensibly built by ‘other participants’ along with the associated scores. These spaceship designs had actually been generated by the experimenter prior to the study using randomly selected tiles, and three out of 12 completed spaceships were presented at random as social sources. The scores for these space ships were also randomly assigned. Participants could choose up to three items from one of the three ships, and these items were automatically added to the participant’s spaceship template in the next phase and could not be removed.”

AR Asocial Risky
Asocial option “their score could markedly increase or decrease if they visited the scrapheap “some of these items may be broken and useless, but some may greatly increase your ship’s score… your score could go up or down”.

Social option: participants were informed heir score could slightly increase or would remain the same “the ships will have the same score as your ship or slightly higher… you will be guaranteed at least the same score as your current ship”.

Social risky (SR)
Asocial option ““all of these items will help your ship to fly and some of them can slightly increase your ships score … you will be guaranteed at least the same score as your current ship”.

Social option ”the ships may have a much worse or much better score than your current ship’s score… your score could go up or down.”

Control: (C)

Asocial “the ships will have the same score as your ship or slightly higher… you will be guaranteed at least the same score as your current ship”.

Social “the ships will have the same score as your ship or slightly higher… you will be guaranteed at least the same score as your current ship”.

3. Results in economics papers typically reproduce the various models in a table. It took some time to adapt to this method but the verbal description is sufficiently careful that a close reading is rewarded and the use of a weighted average fit does reduce the sometimes tedious descriptions of various models.

Therefore the article as written could benefit from yet another rewrite but meets the basic requirements of clarity, references and organizational structure.

Experimental design

The research question clearly fits within the aims and scope of PeerJ. Human and animal studies on risk, social behavior and gender may ultimately provide clear guidance as to which features of humanity are cultural and which stem from more basic evolutionary pressures. Therefore, the goals are clear and address an identified knowledge gap. The procedures are mostly clear although I would need the precise wording given subjects before attempting a replication. The use of deception is within the norms of psychology but not economics. The researchers are psychologists though and therefore conform to the ethics of their field.

Validity of the findings

Unfortunately, I do not trust the findings. Given subjects in an experiment for 20 minutes with little at stake, it may well be that decisions are based on small differences. For example, the description of the social and asocial conditions have one obvious difference - length and complexity. Perhaps the reason both genders prefer the asocial option in the control is because it is simpler. For the social option they have to decide if they believe the scores given spaceships are relevant and which features account for score differences. In the asocial option they have to decide which of 10 items to include. Notice that the social option is social only in the description of information and not any actual social interaction and does not in fact embody any wisdom based on group decision making.

On the other hand, there is a significant interaction effect. Women, but not men, prefer the social information source when the asocial information source is described as being risky. I do not see how this can be attributed to complexity. I suspect there is some other small effect I have missed but I do not know that.

I do trust the result that those told their spaceship is substandard are more willing to take risk. I would at least mention this is consistent with Kahneman and Tversky's prospect theory.

Additional comments

As an economist I find the use of deception difficult to defend. I have tried hard to keep this out of the review but believe you should be aware of the economists' position. Debriefing means subjects eventually learn of the deception and may well participate in a later experiment where they will know not to trust the experimenter. I am not sure why deception is needed here. It seems to be used to save money and time, not inconsiderable motives but not particularly convincing either. If you want non-psychologists to pay attention, you may want to put in a little effort defending why deception is sufficiently useful in this study to outweigh the objection above.

---

## Round 0.2 · accepted · Accept

The three reviewers who reviewed your original submission have now reviewed your revised manuscript and all are in agreement that it is much improved. I am happy to let you know that I am recommending your article for publication in PeerJ. I particularly liked the inclusion of the new figure, which aided clarity.

One reviewer had difficulty viewing your raw data, but as I was able to download the file and open it without issue, I have no concerns regarding that. Additionally, reviewer 1 suggests that you provide more discussion within your manuscript regarding the bounded risk model. I believe that this addition would enhance your article, but is not necessary and I leave that to your discretion.

·

Basic reporting

Errors from the previous version were fixed.

Experimental design

Design is good.

Validity of the findings

Now that the authors, in their rebuttal letter, have clarified the bounded risk model, the findings make more sense. However, I would prefer to see this model be introduced earlier in the manuscript, perhaps in the introduction. This model initially confused both myself and a prior reviewer, suggesting this information needs to be highlighted in a better way in the manuscript.

Reviewer 2 ·

Basic reporting

As before, I am happy with the reporting.

Sorry for being annoying about the raw data file, but it still doesn't display correctly for me, regardless of whether on home or office computer, in Excel or R. I still get values such as 18-25 appearing under "Edu", and Undergraduate Degree appearing under "Country". It would definitely be helpful if the Editor could also check whether it displays OK. Sincere apologies if I am just doing something weird with the file.

Experimental design

As before, no problems to report.

Validity of the findings

I think the handful of new additions to the text help to make the authors' interpretation clearer and more explicit.

·

Basic reporting

See previous review.

Experimental design

See previous review.

Validity of the findings

See previous review.

Additional comments

I have checked the responses to my earlier remarks. Adding the screen shots subjects actually see describing the four fold decision greatly facilitates reading and removes many of my objections.

I was hoping the authors would defend the use of deception in the article, not just the reply to me. It is as I suspected, it saves time and money. For support you could have used Kahnenman - I forget which article right now - but he defends use of unpaid questionnaires because they are faster. He was able to do a new set each week. Made me jealous. Another possible defense is that the number of experiments in a given location is so small that contaminating the subject pool is unlikely.

I recommended minor modifications before and did not demand to see the article again. Glad I saw the screen shots. Seeing more of the program might remove the comments about complexity as well. I believe paper should be accepted.